# Reliability Prediction and FMEA of Loading and Unloading Truss Robot for CNC Punch

**Kaiyue Zhang [1], Zhixin Jia [1,*], Renpeng Bian [1,2], Ketai He [1] and Zhicheng Jia [3]**

[1] School of Mechanical Engineering, University of Science and Technology Beijing, Beijing 100083, China
[2] School of Mechanical and Electrical Engineering, China University of Petroleum (East China), Qingdao 266580, China
[3] Institute of Science and Technology, University of Sanya, Sanya 572100, China
**\*** Correspondence: jiazhixin@me.ustb.edu.cn

**Abstract:** Loading and unloading truss robot for computer numerical control (CNC) punch is widely used in the production of sheet metal parts, as its reliability level is directly related to the quality of sheet metal parts. Especially for the automatic sheet metal production line, it is urgent to predict the reliability of the loading and unloading truss robot for CNC punch. In this paper, a new method for the reliability prediction of the loading and unloading truss robot for CNC punch is proposed. The method uses the component counting method to predict the failure rate and mean time between failures (MTBF) of the electrical control system. Then, according to the MTBF value of the electrical control system, the MTBF value and failure rate of the mechanical system and pneumatic system are calculated by expert scoring method based on fuzzy theory. The MTBF value and failure rate of the loading and unloading truss robot for CNC punch are estimated, and the weak link of the loading and unloading truss robot for CNC punch is obtained. Finally, through the collected fault maintenance data records, the failure modes and effects analysis (FMEA) of the loading and unloading truss robot for CNC punch is carried out, and the reliability prediction method proposed in this paper is validated by the fault location analysis results.

**Keywords:** truss robot; reliability prediction; FMEA; CNC punch; MTBF

## 1. Introduction

In traditional punching machine processing enterprises, the loading and unloading work is repeated, and the labor intensity is large, which occupies a considerable proportion of the total workload. Therefore, the current domestic and foreign market has a strong demand for complete sets of intelligent equipment. For sheet metal processing enterprises, it is urgent to automate the integration of CNC punch, CNC bending machine and other sheet metal processing equipment to greatly improve production efficiency and benefit. Therefore, as an important part of the automatic intelligent production line of sheet metal, the quality and reliability of the loading and unloading truss robot for CNC punch is very important. At present, the reliability level of metal plate automatic processing production line manufactured in China is low, so it is urgent to study the reliability of the loading and unloading truss robot for CNC punch, to improve its reliability level in the automatic production line.

The loading and unloading truss robot for CNC punch belongs to an industrial robot, which can not only be used with CNC punching machine alone, but also can be combined with other machine tools to form an automatic production line. As industrial robots and automated production lines have been gradually popularized in recent years, their reliability research is still relatively weak at present. Guo of Dalian University of Technology designed a loading and unloading truss robot system for CNC lathe. Although finite element analysis and simulation tests were carried out on the system, reliability analysis was not carried out [1]. Although Li from South China University of Technology designed

and studied a small stamping and loading pneumatic manipulator and conducted reliability analysis on its control circuit [2], he only conducted reliability research on the control circuit and the reliability analysis involved was not deep enough. Pang et al. [3] proposed a fuzzy Markov method based on the advantages of fault tree, fuzzy comprehensive evaluation and Markov methods, which can improve the reliability of input fault data and get the updated model which can best reflect the state of the system. Kumar et al. [4] put forward a software reliability prediction algorithm based on ELM and SVM for the problem of low accuracy of fault prediction based on standard machine learning technology, and studied the factors affecting the prediction accuracy. Hao et al. [5] proposed a reliability prediction method with the introduction of interval hierarchy analysis, aiming at the problem that it is difficult to accurately assess the difference degree of reliability level between evaluation objects and similar products when the traditional similarity comparison method is used for reliability prediction of CNC machine tools. Luan et al. [6] studied a reliability fuzzy prediction method based on fuzzy theory in order to overcome the lack of reliability data of existing welding robot systems. Zhang et al. [7] proposed a new reliability prediction method based on "function–motion–action" structural decomposition, aiming at the problem of a large deviation of predicted results due to the large difference between the selected similar products and the evaluation objects when using the traditional similar product method for reliability prediction.

For failure modes and effects analysis (FMEA), as early as 1978, some scholars predicted the reliability level of CNC machine tools under given operating conditions and analyzed different types of failure. In recent years, the theory and application of fault mode analysis are more mature. Shenvi [8] started from the positioning of FMEA in the software life cycle, elaborated the process steps of FMEA and the custom definition that can be used for software development, and gave the analysis results of FMEA. Hu et al. [9] proposed the FMEA method based on functional modeling, gave the analysis process, and verified the FMEA method based on functional modeling. Sun [10] proposed a risk ranking method based on fuzzy evidential reasoning and improved grey correlation degree, aiming at the deficiencies of failure mode and effect analysis (FMEA) in the identification of failure mode risk of crude oil tank of floating production storage and unloading device. Wang et al. [11] conducted reliability analysis on the power and transmission system of HR50 series bending robots. Based on FMEA theory, they used the FMEA analysis method to analyze the failure mode, fault location and fault causes, and found out the weak link of bending robots. Li et al. [12] failed to consider the performance degradation of the solar wing drive mechanism and the randomness and time dependence of load in the traditional mechanical reliability analysis. Based on the uncertainty analysis and fault mode influence analysis of solar wing drive mechanism, the failure criterion is studied, and a time-varying reliability model with high precision and more consistent with the reality is established. Chen et al. [13] proposed a method based on fuzzy data envelopment analysis to determine the weight of factors affecting the fault mode and analyzed the fault mode of CNC machine tools. Yang et al. [14] introduced formal technology into electromechanical system FMEA, proposed an electromechanical system FMEA method based on model detection, and applied the above method to CNC machine tool feed system, successfully identified the system fault caused by the failure of the limit switch through model detection, and verified the feasibility of this method.

Reliability prediction, failure mode and effect analysis have important application value in CNC machine tool and automatic production line reliability research, its value is mainly reflected in the selection of prediction and analysis methods and the accuracy of predicted results. Since reliability and fault analysis plays an important role in tracking and improving the efficiency of machine systems and subsystems [15], it is necessary to consider reliability prediction and FMEA in a comprehensive way. However, in the above literature and research it can be found that, at present, in the relevant reliability studies, most scholars often only carry out reliability prediction or failure mode and impact analysis for a product alone, and do not combine FMEA with predicted results, and there is a lack

of reliability prediction results verification. For the loading and unloading truss robot for CNC punch, there is not a complete set of reliability research methods, let alone the reliability research case of the loading and unloading truss robot for CNC punch. Therefore, it is very important to combine reliability prediction with failure mode and impact analysis and verify the predicted results.

Based on the loading and unloading truss robot for CNC punch in the automatic sheet metal production line, this paper studies its reliability and proposes a new reliability prediction method. By establishing the reliability model of the loading and unloading truss robot for CNC punch, using the component counting method and fuzzy theory to predict the failure rate and mean time between failures of each system, to find out the weak link, and then according to the on-site fault data of the loading and unloading truss robot for CNC punch FMEA analysis, verify and determine the weak link. Finally, the reliability requirements of each subsystem of the loading and unloading truss robot for CNC punch are clarified through reliability distribution.

## 2. Materials and Methods

### 2.1. Reliability Prediction

Reliability prediction is to predict the possible reliability index of the equipment based on the functional structure of the equipment, the expected working environment and the reliability data of the parts or components of the equipment [16]. Reliability prediction of products can find the weak links of the system in advance, and through repeated reliability prediction, more appropriate design schemes can be obtained. On the contrary, if the reliability prediction is not carried out, the reliability index will not reach the standard after the successful design and manufacturing of the product due to the failure to implement the necessary reliability work, resulting in the loss of economic benefits of the enterprise. Therefore, it is of great significance to predict the reliability of the loading and unloading truss robot for CNC punch.

For the loading and unloading truss robot for CNC punch, the main purpose and significance of reliability prediction are as follows:

(1) Evaluate whether each system of the loading and unloading truss robot for CNC punch can reach the specified reliability index;

(2) Lay the foundation for reliability distribution of the loading and unloading truss robot for CNC punch;

(3) According to the reliability prediction results, the weak links of the loading and unloading truss robot for CNC punch are discovered in advance to reduce the research and development cost and time;

(4) Develop preventive maintenance plan, parts reserve and update plan for the loading and unloading truss robot for CNC punch.

### 2.2. Failure Modes and Effects Analysis

Failure modes and effects analysis [17] (FMEA) is a systematic reliability analysis theory. FMEA is widely used in various links of product design, development and quality improvement. By analyzing the fault location, fault mode and fault cause of the product, the weak link of the product can be found out, and then technical improvement can be made to address the weak link, to ultimately improve the product quality and reliability [18,19]. Under normal circumstances, FMEA includes fault location, fault mode, fault cause and fault impact analysis, and finally FEMA output in table form.

### 2.3. Combination of Reliability Prediction and FMEA

In this paper, a reliability prediction method of the loading and unloading truss robot for CNC punch is proposed by using the component counting method and fuzzy theory. The specific idea is as follows: firstly, the failure rate of the existing components is used to predict the failure rate and MTBF value of the electrical control system by the counting method of components. Secondly, based on the predicted results of the electrical control

system, the failure rate and MTBF values of the mechanical system and pneumatic system are predicted by using the fuzzy theory. Finally, the failure rate and MTBF value of the loading and unloading truss robot for CNC punch are predicted, and the weak link is analyzed.

Then, the failure frequency, failure mode and main failure reasons of the loading and unloading truss robot for CNC punch system are calculated through FMEA analysis of the collected the loading and unloading truss robot for CNC punch fault data, and the weak link is obtained, in order to verify the effectiveness of the reliability prediction method proposed in this paper and the accuracy of the predicted results. The detailed procedure flow chart is shown in Figure 1.

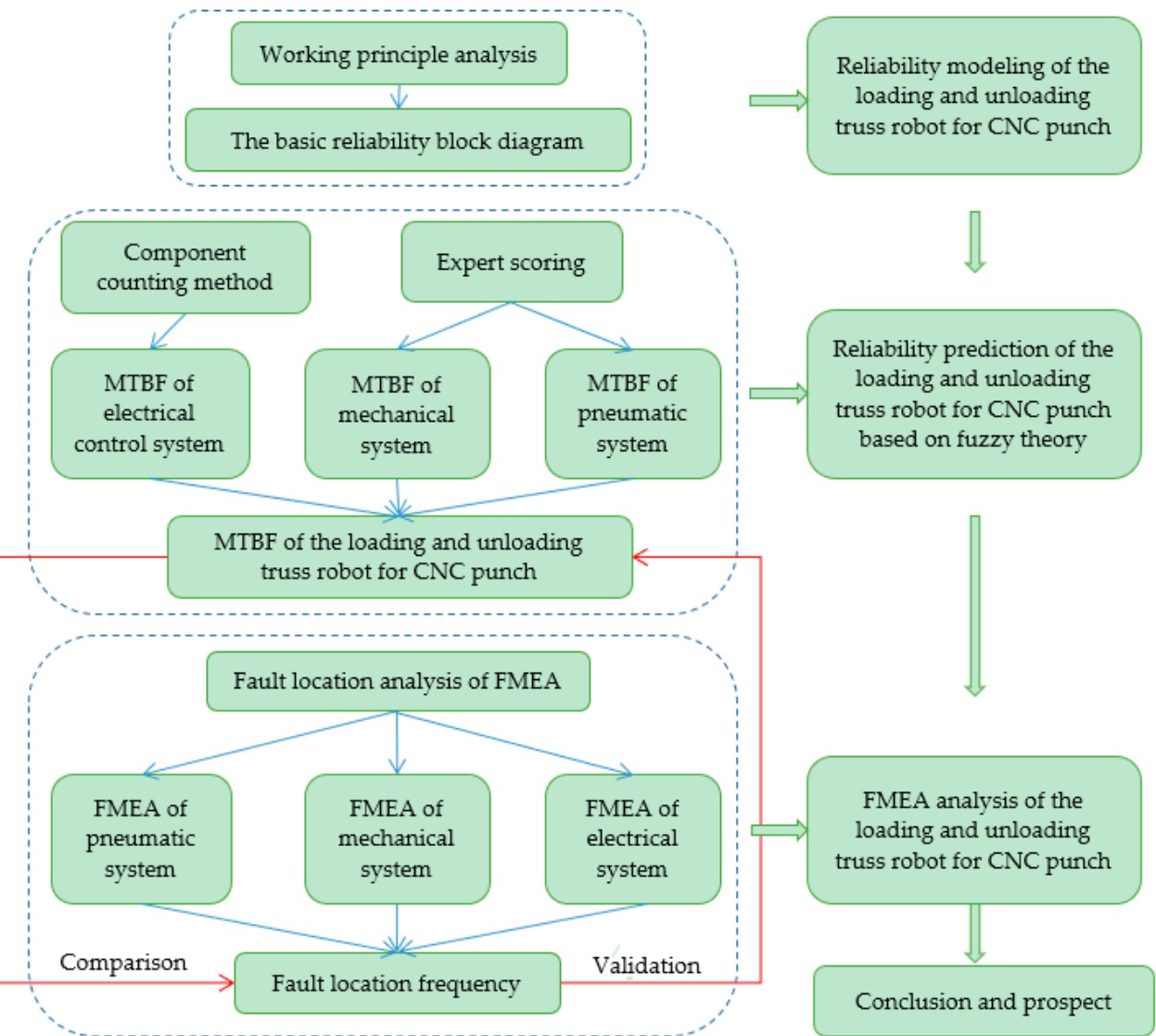

**Figure 1.** Flow diagram of reliability prediction and FMEA analysis of the loading and unloading truss robot for CNC punch.

## 3. Results

In this paper, taking a certain type of the loading and unloading truss robot for CNC punch as an example, the reliability prediction is carried out by using the component counting method and the fuzzy expert scoring method to find out the weak links of the truss robot, and the reliability prediction results are verified by FMEA analysis of the loading and unloading truss robot for CNC punch.

### 3.1. Establishment of Reliability Model

The loading and unloading truss robot for CNC punch is a part of the automatic intelligent production line of sheet metal. The design of integrating X and U axis in a single beam is adopted. Its main components include beam, clamp, end tooling, hydraulic lifting platform, raw material car, electric cabinet, pneumatic system and suction cup, etc. Its structure diagram is shown in Figure 2.

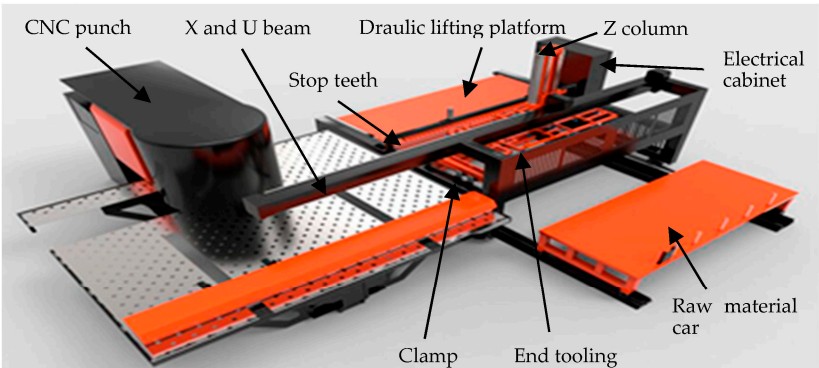

**Figure 2.** Schematic diagram of the structure of the loading and unloading truss robot for CNC punch.

The working principle of the loading and unloading truss robot for CNC punch is as follows: the movement of the raw material car is controlled, the sheet material to be processed is sent to the set position, and then the operation of the X and Z axis motor is controlled by the electric control system. The loading mechanism moves in the direction of X axis and Z axis at the same time and moves to the raw material car. The sheet material is sucked up with the sucker by the pneumatic control system, and then the sheet material is sent to the CNC punch for processing, and the loading is completed. At the same time, the clamp is controlled, moving in the direction of U-axis by the electric control system, the processed sheet material is picked up and sent to the moving brush platform, and then the finished sheet material on the brush platform is blocked by the stop teeth on the hydraulic lifting platform to complete the blanking. Loading and unloading are carried out at the same time without interference. According to the component relationship of the loading and unloading truss robot for CNC punch, it is divided into mechanical system (beam, clamp, end tool, hydraulic lifting platform, raw material car, etc.), pneumatic system (suction cup) and electrical control system (electric cabinet).

Then, the basic reliability block diagram of the loading and unloading truss robot of the CNC punch is established, as shown in Figure 3. The basic reliability block diagram of the loading and unloading truss robot for CNC punch includes the mechanical system block diagram (M), the pneumatic system block diagram (P) and the electrical control system (E), in which the letter behind each system indicates the code of the system. As long as one system fails, the loading and unloading truss robot for CNC punch will fail. Therefore, the reliability model of each system is a series model.

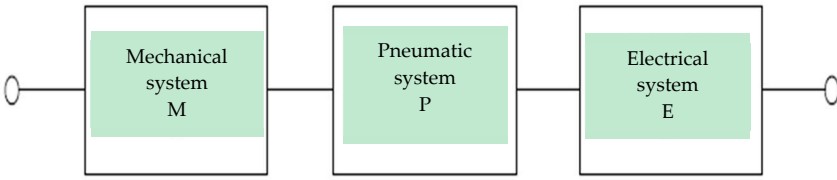

**Figure 3.** The basic reliability block diagram of the loading and unloading truss robot for CNC punch.

*3.2. Reliability Prediction*

3.2.1. Reliability Prediction of Electrical Control System

Firstly, the failure rate is selected as the reliability prediction index, and the component counting method is used to predict the electrical control system. The first step is to collect the name, model, quantity and failure rate of each component, and then calculate the cumulative failure rate. The failure rate collection of electronic components is based on GB/T 37963-2019 "Electronic Equipment Reliability Prediction Model and Data Manual", GJB 299-87 "Electronic Equipment Reliability Prediction Manual" and the component failure rate provided by the supplier. The reliability prediction model is a series system, and the failure rate prediction model of the component counting method [20] is

$$\lambda_{GS} = \sum_{i=1}^{n} N_i \lambda_{Gi} \qquad (1)$$

In the equation: $\lambda_{GS}$ is the total failure rate of the system; $N_i$ is the number of the *i*th component; $\lambda_{Gi}$ is the failure rate of the *i*th component; n is the number of types of system components. According to the above reliability prediction model, the components of the electrical control system are counted and calculated, and the reliability prediction table is shown in Table 1.

**Table 1.** Reliability prediction table.

| Serial Number | Name | Quantity | Failure Rate (1/h) | Cumulative Failure Rate (1/h) |
|---|---|---|---|---|
| 1 | Sensor | 34 | $3.858 \times 10^{-6}$ | $1.312 \times 10^{-4}$ |
| 2 | Connector | 500 | $7.890 \times 10^{-8}$ | $3.945 \times 10^{-5}$ |
| 3 | Digital module | 6 | $5.787 \times 10^{-6}$ | $3.472 \times 10^{-5}$ |
| 4 | Circuit breaker | 7 | $2.894 \times 10^{-6}$ | $2.026 \times 10^{-5}$ |
| 5 | AC contactor | 3 | $6.705 \times 10^{-6}$ | $2.012 \times 10^{-5}$ |
| 6 | Light curtain | 2 | $5.787 \times 10^{-6}$ | $1.157 \times 10^{-5}$ |
| 7 | Intermediate relay | 5 | $2.251 \times 10^{-6}$ | $1.126 \times 10^{-5}$ |
| 8 | Touch screen | 1 | $9.645 \times 10^{-6}$ | $9.645 \times 10^{-6}$ |
| 9 | PLC | 1 | $8.267 \times 10^{-6}$ | $8.267 \times 10^{-6}$ |
| 10 | Communication module | 1 | $7.716 \times 10^{-6}$ | $7.716 \times 10^{-6}$ |
| 11 | Switch | 1 | $6.430 \times 10^{-6}$ | $6.430 \times 10^{-6}$ |
| 12 | Analog module | 1 | $5.787 \times 10^{-6}$ | $5.787 \times 10^{-6}$ |
| 13 | Servo drive | 3 | $9.873 \times 10^{-7}$ | $2.962 \times 10^{-6}$ |
| 14 | Switching power supply | 2 | $1.250 \times 10^{-6}$ | $2.500 \times 10^{-6}$ |
| 15 | Tricolor lights | 1 | $9.690 \times 10^{-7}$ | $9.690 \times 10^{-7}$ |
| 16 | Breakout connector | 8 | $6.903 \times 10^{-8}$ | $5.522 \times 10^{-7}$ |
| 17 | Isolating switch | 1 | $3.041 \times 10^{-7}$ | $3.041 \times 10^{-7}$ |
| 18 | Operating handle | 1 | $3.041 \times 10^{-7}$ | $3.041 \times 10^{-7}$ |
| 19 | Switch | 1 | $3.041 \times 10^{-7}$ | $3.041 \times 10^{-7}$ |
| 20 | Button | 3 | $7.180 \times 10^{-8}$ | $2.154 \times 10^{-7}$ |
| 21 | Frequency converter | 2 | $2.355 \times 10^{-8}$ | $4.709 \times 10^{-8}$ |
| | Statistics | | | $3.146 \times 10^{-4}$ |

The text continues here (Figure 2 and Table 2). It can be concluded from Table 1 that the failure rate of the electrical control system of the loading and unloading truss robot of the CNC punch is $3.146 \times 10^{-4}$ (1/h), and *MTBF* = 3178.640 h, calculated from the equation

$$MTBF = 1/\lambda.$$

**Table 2.** Fuzzy number of rating factors and relative.

| Evaluation Factor Language Fuzzy Variable | Level | Fuzzy Number |
|---|---|---|
| Exceptionally High (VH) | 9–10 | (8, 9, 10), (9, 10, 10) |
| High (H) | 7–8 | (6, 7, 8), (7, 8, 9) |
| Medium (M) | 5–6 | (4, 5, 6), (5, 6, 7) |
| Low | 3–4 | (2, 3, 4), (3, 4, 5) |
| Extra Low (VL) | 1–2 | (1, 1, 2), (1, 2, 3) |

### 3.2.2. Reliability Prediction of Mechanical System and Pneumatic System

The mechanical system and pneumatic system of the loading and unloading truss robot for CNC punch include a large number of parts and components, and the failure mechanisms of the parts and components of the system are different. The expert grading method based on fuzzy theory [21] is used to predict the reliability of the mechanical system and pneumatic system of the loading and unloading truss robot for CNC punch.

According to the fuzzy reliability prediction model [22], the loading and unloading truss robot of CNC punch is predicted, the specific implementation methods are as follows: the mechanical system (M), pneumatic system (P) and electrical control system (E) were scored by five expert group members based on fuzzy theory according to complexity, technical level, working time and environmental conditions. The grades of the four evaluation factors are divided into 10 grades, and then fuzzy theory is introduced to deal with the continuous triangular fuzzy numbers of the linguistic variables. The grades and relative importance fuzzy numbers of the four evaluation factors are shown in Table 2.

Now five experienced loading and unloading truss robot of CNC punch designers and experts are invited to independently score the corresponding evaluation factors. Under the four scoring factors of complexity, technical level, working time and environmental conditions, the expert scoring results table of each system is shown in Table 3.

**Table 3.** Each system expert score results table.

| Evaluation Factors | Expert | M | P | E |
|---|---|---|---|---|
| Complexity | Z1 | (8, 9, 10) | (1, 2, 3) | (6, 7, 8) |
| | Z2 | (7, 8, 9) | (2, 3, 4) | (7, 8, 9) |
| | Z3 | (8, 9, 10) | (1, 2, 3) | (6, 7, 8) |
| | Z4 | (8, 9, 10) | (3, 4, 5) | (8, 9, 10) |
| | Z5 | (7, 8, 9) | (2, 3, 4) | (6, 7, 8) |
| Technical level | Z1 | (1, 2, 3) | (2, 3, 4) | (5, 6, 7) |
| | Z2 | (1, 1, 2) | (3, 4, 5) | (4, 5, 6) |
| | Z3 | (1, 1, 2) | (2, 3, 4) | (6, 7, 8) |
| | Z4 | (2, 3, 4) | (2, 3, 4) | (5, 6, 7) |
| | Z5 | (1, 1, 2) | (4, 5, 6) | (7, 8, 9) |
| Working hours | Z1 | (5, 6, 7) | (8, 9, 10) | (8, 9, 10) |
| | Z2 | (7, 8, 9) | (9, 10, 10) | (9, 10, 10) |
| | Z3 | (6, 7, 8) | (7, 8, 9) | (9, 10, 10) |
| | Z4 | (6, 7, 8) | (8, 9, 10) | (8, 9, 10) |
| | Z5 | (7, 8, 9) | (8, 9, 10) | (8, 9, 10) |
| Environmental conditions | Z1 | (5, 6, 7) | (6, 7, 8) | (1, 1, 2) |
| | Z2 | (2, 3, 4) | (5, 6, 7) | (1, 1, 2) |
| | Z3 | (5, 6, 7) | (7, 8, 9) | (1, 2, 3) |
| | Z4 | (4, 5, 6) | (7, 8, 9) | (1, 2, 3) |
| | Z5 | (4, 5, 6) | (6, 7, 8) | (1, 1, 2) |

As can be seen from Table 3, different experts give inconsistent scores to the same index. Therefore, expert scoring is subjective to a certain extent. To eliminate the influence of experts' personal opinions, the analytic hierarchy process is used to calculate the weight of each expert, and then, according to the weight of each expert, the evaluation value of each expert is calculated based on fuzzy theory. The specific methods for determining expert weight are as follows:

In the hierarchical structure model, the top layer is the target layer, the second layer is the factor layer, the third layer is the sub-factor layer and the bottom layer is the expert layer. The objective layer is expert weight, the factor layer constitutes factor set $B = \{B_1, B_2, B_3, B_4\}$ = {professional title, educational background, seniority, age}, and then subdivide each factor to form a sub-factor layer $\{B_{ij}\}(j = 1, 2, \ldots, m)$, a set of factors of expert layer $Z = \{Z_1, Z_2, Z_3, Z_4\}$. The hierarchical model for each expert weight is shown in Figure 4.

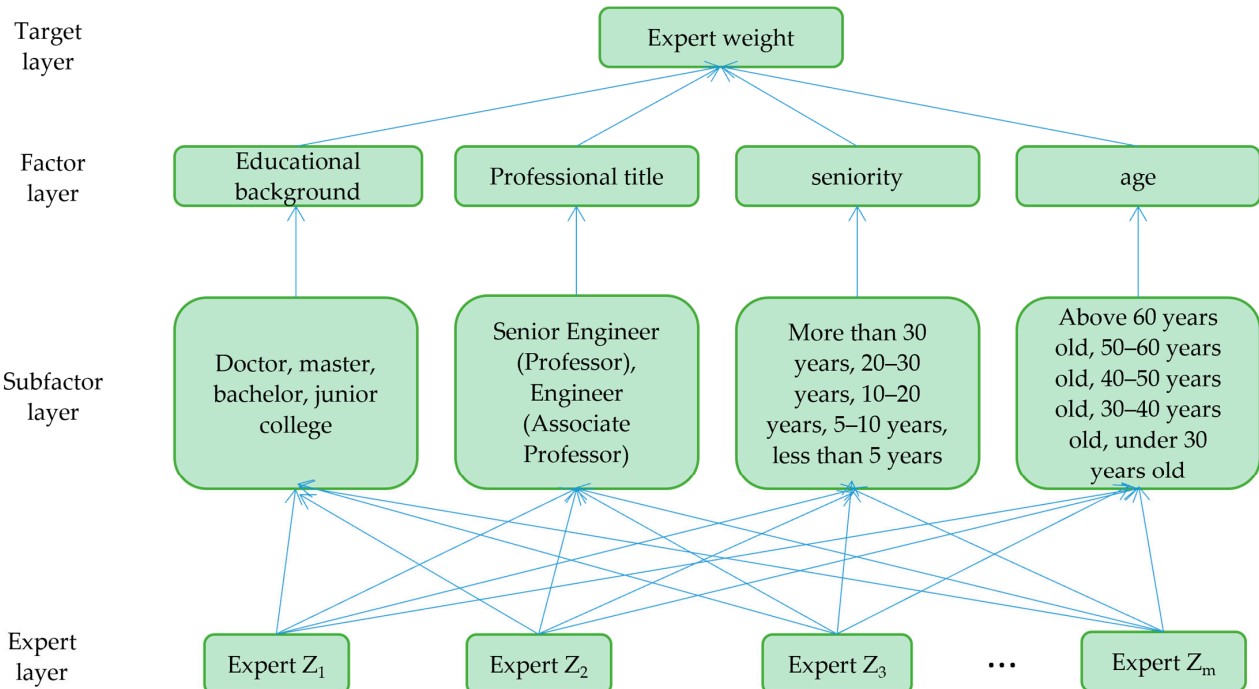

**Figure 4.** Expert weight hierarchy model.

In the expert weight hierarchy model shown in Figure 4, the weight coefficient of the factor layer of the second layer is set as $W_i$ = {$w_1$, $w_2$, $w_3$, $w_4$} = {1, 2, 3, 4}, and the weight coefficient of the sub-factor layer is set as $W_{ij}$ = {1, 2, 3, 4, 5}, and then the corresponding weight is calculated according to the personal data of the experts. According to the basic information of the five experts, the weight of each expert is 0.23, 0.19, 0.14, 0.13, 0.31. The weights of scoring factors and experts are shown in Table 4.

**Table 4.** Expert weight and evaluation factor weight table.

| Expert | Expert Weight | Complexity Weight | Technical Level Weight | Work Time Weight | Environmental Condition Weight |
|---|---|---|---|---|---|
| Z1 | 0.23 | (3, 4, 5) | (2, 3, 4) | (1, 1, 2) | (1, 1, 2) |
| Z2 | 0.19 | (2, 3, 4) | (2, 3, 4) | (2, 3, 4) | (1, 2, 3) |
| Z3 | 0.14 | (4, 5, 6) | (3, 4, 5) | (1, 2, 3) | (1, 2, 3) |
| Z4 | 0.13 | (2, 3, 4) | (2, 3, 4) | (2, 3, 4) | (2, 3, 4) |
| Z5 | 0.31 | (3, 4, 5) | (3, 4, 5) | (1, 2, 3) | (1, 1, 2) |

The calculation process of evaluation factors and weight analysis of CNC punch loading and unloading truss robot is as follows:

After $m$ ($m \geq 5$) experts score $n$ systems, the scoring coefficients ($\omega_i$) of each system of the loading and unloading truss robot of CNC punch are calculated.

Firstly, the fuzzy rating levels of complexity, technical level, working time and environmental conditions for the $i$th system are, respectively, set as:

$$\widetilde{R}_{ij}^1 = (R_{ijL}^1, R_{ijM}^1, R_{ijU}^1), \ \widetilde{R}_{ij}^2 = (R_{ijL}^2, R_{ijM}^2, R_{ijU}^2), \ \widetilde{R}_{ij}^3 = (R_{ijL}^3, R_{ijM}^3, R_{ijU}^3),$$
$$\widetilde{R}_{ij}^4 = (R_{ijL}^4, R_{ijM}^4, R_{ijU}^4)(I = 1, 2, \ldots, n; \ j = 1, 2, \ldots, m).$$

Then, the fuzzy weights of the four evaluation factors given by the $j$th expert were, respectively, set as:

$$\widetilde{\omega}_j^1 = (\omega_{jL}^1, \omega_{jM}^1, \omega_{jU}^1), \ \widetilde{\omega}_j^2 = (\omega_{jL}^2, \omega_{jM}^2, \omega_{jU}^2), \ \widetilde{\omega}_j^3 = (\omega_{jL}^3, \omega_{jM}^3, \omega_{jU}^3),$$
$$\widetilde{\omega}_j^4 = (\omega_{jL}^4, \omega_{jM}^4, \omega_{jU}^4) \ (j = 1, 2, \ldots, m).$$

Set the relative weight of the *m*th expert to $h_j\left(\sum\limits_{j=1}^{m} h_j = 1, h_j > 0, j = 1, \cdots, m\right)$. Then, the fuzzy scoring coefficient solving model of each system of the loading and unloading truss robot of CNC punch is

$$\widetilde{R}_i^1 = \sum_{j=1}^{m} h_j \widetilde{R}_{ij}^1 = (\sum_{j=1}^{m} h_j \widetilde{R}_{ijL}^1, \sum_{j=1}^{m} h_j \widetilde{R}_{ijM}^1, \sum_{j=1}^{m} h_j \widetilde{R}_{ijU}^1), i = 1, 2, \cdots, n \tag{2}$$

$$\widetilde{R}_i^2 = \sum_{j=1}^{m} h_j \widetilde{R}_{ij}^2 = (\sum_{j=1}^{m} h_j \widetilde{R}_{ijL}^2, \sum_{j=1}^{m} h_j \widetilde{R}_{ijM}^2, \sum_{j=1}^{m} h_j \widetilde{R}_{ijU}^2), i = 1, 2, \cdots, n \tag{3}$$

$$\widetilde{R}_i^3 = \sum_{j=1}^{m} h_j \widetilde{R}_{ij}^3 = (\sum_{j=1}^{m} h_j \widetilde{R}_{ijL}^3, \sum_{j=1}^{m} h_j \widetilde{R}_{ijM}^3, \sum_{j=1}^{m} h_j \widetilde{R}_{ijU}^3), i = 1, 2, \cdots, n \tag{4}$$

$$\widetilde{R}_i^4 = \sum_{j=1}^{m} h_j \widetilde{R}_{ij}^4 = (\sum_{j=1}^{m} h_j \widetilde{R}_{ijL}^4, \sum_{j=1}^{m} h_j \widetilde{R}_{ijM}^4, \sum_{j=1}^{m} h_j \widetilde{R}_{ijU}^4), i = 1, 2, \cdots, n \tag{5}$$

$$\widetilde{\omega}^1 = \sum_{j=1}^{m} h_j \widetilde{\omega}_j^1 = (\sum_{j=1}^{m} h_j \widetilde{\omega}_{jL}^1, \sum_{j=1}^{m} h_j \widetilde{\omega}_{jM}^1, \sum_{j=1}^{m} h_j \widetilde{\omega}_{jU}^1), j = 1, 2, \cdots, m \tag{6}$$

$$\widetilde{\omega}^2 = \sum_{j=1}^{m} h_j \widetilde{\omega}_j^2 = (\sum_{j=1}^{m} h_j \widetilde{\omega}_{jL}^2, \sum_{j=1}^{m} h_j \widetilde{\omega}_{jM}^2, \sum_{j=1}^{m} h_j \widetilde{\omega}_{jU}^2), j = 1, 2, \cdots, m \tag{7}$$

$$\widetilde{\omega}^3 = \sum_{j=1}^{m} h_j \widetilde{\omega}_j^3 = (\sum_{j=1}^{m} h_j \widetilde{\omega}_{jL}^3, \sum_{j=1}^{m} h_j \widetilde{\omega}_{jM}^3, \sum_{j=1}^{m} h_j \widetilde{\omega}_{jU}^3), j = 1, 2, \cdots, m \tag{8}$$

$$\widetilde{\omega}^4 = \sum_{j=1}^{m} h_j \widetilde{\omega}_j^4 = (\sum_{j=1}^{m} h_j \widetilde{\omega}_{jL}^4, \sum_{j=1}^{m} h_j \widetilde{\omega}_{jM}^4, \sum_{j=1}^{m} h_j \widetilde{\omega}_{jU}^4), j = 1, 2, \cdots, m \tag{9}$$

According to Equations (2)–(5), the overall fuzzy evaluation of influencing factors of each system is:

$$\widetilde{R}_i^1 = (R_{iL}^1, R_{iM}^1, R_{iU}^1), \ \widetilde{R}_i^2 = (R_{iL}^2, R_{iM}^2, R_{iU}^2), \ \widetilde{R}_i^3 = (R_{iL}^3, R_{iM}^3, R_{iU}^3),$$
$$\widetilde{R}_i^4 = (R_{iL}^4, R_{iM}^4, R_{iU}^4) \ (i = 1, 2, \ldots, n)$$

According to Equations (6)–(9), the fuzzy weight of each evaluation factor given by the experts is:

$$\widetilde{\omega}^1 = (\omega_L^1, \omega_M^1, \omega_U^1), \ \widetilde{\omega}^2 = (\omega_L^2, \omega_M^2, \omega_U^2), \ \widetilde{\omega}^3 = (\omega_L^3, \omega_M^3, \omega_U^3),$$
$$\widetilde{\omega}^4 = (\omega_L^4, \omega_M^4, \omega_U^4)$$

According to the fuzzy weight arithmetic mean equation, the process of calculating the fuzzy scoring coefficient is derived:

$$F\omega_i = (\widetilde{R}_i^1)^{\frac{\widetilde{\omega}^1}{\widetilde{\omega}^1 + \widetilde{\omega}^2 + \widetilde{\omega}^3 + \widetilde{\omega}^4}} \times (\widetilde{R}_i^1)^{\frac{\widetilde{\omega}^1}{\widetilde{\omega}^1 + \widetilde{\omega}^2 + \widetilde{\omega}^3 + \widetilde{\omega}^4}} \times (\widetilde{R}_i^2)^{\frac{\widetilde{\omega}^2}{\widetilde{\omega}^1 + \widetilde{\omega}^2 + \widetilde{\omega}^3 + \widetilde{\omega}^4}}$$
$$\times (\widetilde{R}_i^3)^{\frac{\widetilde{\omega}^3}{\widetilde{\omega}^1 + \widetilde{\omega}^2 + \widetilde{\omega}^3 + \widetilde{\omega}^4}} \times (\widetilde{R}_i^4)^{\frac{\widetilde{\omega}^4}{\widetilde{\omega}^1 + \widetilde{\omega}^2 + \widetilde{\omega}^3 + \widetilde{\omega}^4}} \tag{10}$$

Taking logarithm of Equation (10), the equation can be derived:

$$\ln(F\omega_i) = \frac{\widetilde{\omega}^1}{\widetilde{\omega}^1 + \widetilde{\omega}^2 + \widetilde{\omega}^3 + \widetilde{\omega}^4} \ln(\widetilde{R}_i^1) + \frac{\widetilde{\omega}^2}{\widetilde{\omega}^1 + \widetilde{\omega}^2 + \widetilde{\omega}^3 + \widetilde{\omega}^4} \ln(\widetilde{R}_i^2)$$
$$+ \frac{\widetilde{\omega}^3}{\widetilde{\omega}^1 + \widetilde{\omega}^2 + \widetilde{\omega}^3 + \widetilde{\omega}^4} \ln(\widetilde{R}_i^3) + \frac{\widetilde{\omega}^4}{\widetilde{\omega}^1 + \widetilde{\omega}^2 + \widetilde{\omega}^3 + \widetilde{\omega}^4} \ln(\widetilde{R}_i^4) \tag{11}$$

$$z_1 = \ln(F\omega_i)_\lambda^L, z_2 = \ln(F\omega_i)_\lambda^U$$

It can be known from Equation (11):

$$
\begin{aligned}
Minz_1 &= \mu_1 \ln\left(R_i^1\right)_\lambda^L + \mu_2 \ln\left(R_i^2\right)_\lambda^L + \mu_3 \ln\left(R_i^3\right)_\lambda^L + \mu_4 \ln\left(R_i^4\right)_\lambda^L \\
\text{s.t. } \mu_1 &= \left(\frac{\widetilde{\omega}^1}{\widetilde{\omega}^1 + \widetilde{\omega}^2 + \widetilde{\omega}^3 + \widetilde{\omega}^4}\right)_\lambda^L, \ \mu_2 = \left(\frac{\widetilde{\omega}^2}{\widetilde{\omega}^1 + \widetilde{\omega}^2 + \widetilde{\omega}^3 + \widetilde{\omega}^4}\right)_\lambda^L \\
\mu_3 &= \left(\frac{\widetilde{\omega}^3}{\widetilde{\omega}^1 + \widetilde{\omega}^2 + \widetilde{\omega}^3 + \widetilde{\omega}^4}\right)_\lambda^L, \ \mu_4 = \left(\frac{\widetilde{\omega}^4}{\widetilde{\omega}^1 + \widetilde{\omega}^2 + \widetilde{\omega}^3 + \widetilde{\omega}^4}\right)_\lambda^L
\end{aligned}
\tag{12}
$$

$$
\begin{aligned}
Maxz_2 &= \mu_1 \ln\left(R_i^1\right)_\lambda^U + \mu_2 \ln\left(R_i^2\right)_\lambda^U + \mu_3 \ln\left(R_i^3\right)_\lambda^U + \mu_4 \ln\left(R_i^4\right)_\lambda^U \\
\text{s.t. } \mu_1 &= \left(\frac{\widetilde{\omega}^1}{\widetilde{\omega}^1 + \widetilde{\omega}^2 + \widetilde{\omega}^3 + \widetilde{\omega}^4}\right)_\lambda^U, \ \mu_2 = \left(\frac{\widetilde{\omega}^2}{\widetilde{\omega}^1 + \widetilde{\omega}^2 + \widetilde{\omega}^3 + \widetilde{\omega}^4}\right)_\lambda^U \\
\mu_3 &= \left(\frac{\widetilde{\omega}^3}{\widetilde{\omega}^1 + \widetilde{\omega}^2 + \widetilde{\omega}^3 + \widetilde{\omega}^4}\right)_\lambda^U, \ \mu_4 = \left(\frac{\widetilde{\omega}^4}{\widetilde{\omega}^1 + \widetilde{\omega}^2 + \widetilde{\omega}^3 + \widetilde{\omega}^4}\right)_\lambda^U
\end{aligned}
\tag{13}
$$

$\left[\left(R_i^1\right)_\lambda^L \text{fl}\left(R_i^1\right)_\lambda^U\right]$ —$\lambda$ cut set of global fuzzy evaluation of system complexity $\widetilde{R}_i^1$;

$\left[\left(R_i^2\right)_\lambda^L \text{fl}\left(R_i^2\right)_\lambda^U\right]$ —$\lambda$ cut set of global fuzzy evaluation of system technical level $\widetilde{R}_i^2$;

$\left[\left(R_i^3\right)_\lambda^L \text{fl}\left(R_i^3\right)_\lambda^U\right]$ —$\lambda$ cut set of global fuzzy evaluation of system working time $\widetilde{R}_i^3$;

$\left[\left(R_i^4\right)_\lambda^L \text{fl}\left(R_i^4\right)_\lambda^U\right]$ —$\lambda$ cut set of global fuzzy evaluation of system environmental condition $\widetilde{R}_i^4$;

$\left[\left(\omega^1\right)_\lambda^L \text{fl}\left(\omega^1\right)_\lambda^U\right]$ —$\lambda$ cut set of complexity fuzzy weights given by experts $\widetilde{\omega}^1$;

$\left[\left(\omega^2\right)_\lambda^L \text{fl}\left(\omega^2\right)_\lambda^U\right]$ —$\lambda$ cut set of fuzzy weights of technical level given by experts $\widetilde{\omega}^2$;

$\left[\left(\omega^3\right)_\lambda^L \text{fl}\left(\omega^3\right)_\lambda^U\right]$ —$\lambda$ cut set of fuzzy weight of working time given by experts $\widetilde{\omega}^3$;

$\left[\left(\omega^4\right)_\lambda^L \text{fl}\left(\omega^4\right)_\lambda^U\right]$ —$\lambda$ cut-off set of fuzzy weights of environmental conditions given by experts $\widetilde{\omega}^4$.

According to the fuzzy geometric weighted average model, and set $Z_1^*$ and $Z_2^*$ as the best objective function values of Equations (12) and (13), the equation $(F\omega_i)_\lambda^L = \exp\left(z_1^*\right)$ and $(F\omega_i)_\lambda^U = \exp\left(z_2^*\right)$ can be obtained. Then, by setting different $\lambda$ values, different $\lambda$ cut sets can be obtained:

$$
F\omega_i = U_\lambda \lambda \cdot \left[(F\omega_i)_\lambda^L \text{fl}(F\omega_i)_\lambda^U\right] \text{fl} 0 < \lambda \le 1
\tag{14}
$$

According to Equations (2)–(9), the relevant data in Tables 3 and 4 are calculated, and the evaluation factors and weight analysis table of the loading and unloading truss robot of CNC punch is obtained, as shown in Table 5.

**Table 5.** Evaluation factors and weight analysis table of the loading and unloading truss robot for CNC punch.

| System | Complexity | Technical Level | Working Hours | Environmental Conditions |
|---|---|---|---|---|
| M | (7.50, 8.50, 9.50) | (1.13, 1.49, 2.49) | (6.27, 7.27, 8.27) | (3.99, 4.99, 5.99) |
| P | (1.76, 2.76, 3.76) | (2.81, 3.81, 4.81) | (8.05, 9.05, 9.86) | (6.08, 7.08, 8.08) |
| E | (6.45, 7.45, 8.45) | (5.57, 6.57, 7.57) | (8.33, 9.33, 10.00) | (1.00, 1.27, 2.27) |
| Factor weight | (2.82, 3.82, 4.82) | (2.45, 3.45, 4.45) | (1.32, 2.09, 3.09) | (1.13, 1.59, 2.59) |

Set the value of $\lambda$, take $\nabla \lambda_i = \frac{1}{5}, i = 1, 2, 3, 4, 5$. Based on this value, $\lambda$ cut sets of the three system evaluation factors of the loading and unloading truss robot of CNC punch with 6 arithmetic sequences are established, as shown in Table 6.

**Table 6.** $\lambda$ cut set of each system evaluation factor.

| System | Cut Set | Complexity | Technical Level | Working Hours | Environmental Conditions |
|---|---|---|---|---|---|
| M | 0 | (7.50, 9.50) | (1.13, 2.49) | (6.27, 8.27) | (3.99, 5.99) |
| | 0.2 | (7.70, 9.30) | (1.20, 2.29) | (6.47, 8.07) | (4.19, 5.79) |
| | 0.4 | (7.90, 9.10) | (1.27, 2.09) | (6.67, 7.87) | (4.39, 5.59) |
| | 0.6 | (8.10, 8.90) | (1.35, 1.89) | (6.87, 7.67) | (4.59, 5.39) |
| | 0.8 | (8.30, 8.70) | (1.42, 1.69) | (7.07, 7.47) | (4.79, 5.19) |
| | 1 | (8.50, 8.50) | (1.49, 1.49) | (7.27, 7.27) | (4.99, 4.99) |
| P | 0 | (1.76, 3.76) | (2.81, 4.81) | (8.05, 9.86) | (6.08, 8.08) |
| | 0.2 | (1.96, 3.56) | (3.01, 4.61) | (8.25, 9.70) | (6.28, 7.88) |
| | 0.4 | (2.16, 3.36) | (3.21, 4.41) | (8.45, 9.54) | (6.48, 7.68) |
| | 0.6 | (2.36, 3.16) | (3.41, 4.21) | (8.65, 9.37) | (6.68, 7.48) |
| | 0.8 | (2.56, 2.96) | (3.61, 4.01) | (8.85, 9.21) | (6.88, 7.28) |
| | 1 | (2.76, 2.76) | (3.81, 3.81) | (9.05, 9.05) | (7.08, 7.08) |
| E | 0 | (6.45, 8.45) | (5.57, 7.57) | (8.33, 10.00) | (1.00, 2.27) |
| | 0.2 | (6.65, 8.25) | (5.77, 7.37) | (8.53, 9.87) | (1.05, 2.07) |
| | 0.4 | (6.85, 8.05) | (5,97, 7,17) | (8.73, 9.73) | (1.11, 1.87) |
| | 0.6 | (7.05, 7.85) | (6.17, 6.97) | (8.93, 9.60) | (1.16, 1.67) |
| | 0.8 | (7.25, 7.65) | (6.37, 6.77) | (9.13, 9.46) | (1.22, 1.47) |
| | 1 | (7.45, 7.45) | (6.57, 6.57) | (9.33, 9.33) | (1.27, 1.27) |

Then, according to the operation theorem of triangular fuzzy function and the relevant data in Table 5, the fuzzy weight $\lambda$ cut set is calculated and established, as shown in Table 7.

**Table 7.** Fuzzy weight $\lambda$ cut set of evaluation factors for loading and unloading truss robot for CNC punch.

| Cut Set | $\dfrac{\widetilde{\omega}^1}{\widetilde{\omega}^1+\widetilde{\omega}^2+\widetilde{\omega}^3+\widetilde{\omega}^4}$ | $\dfrac{\widetilde{\omega}^2}{\widetilde{\omega}^1+\widetilde{\omega}^2+\widetilde{\omega}^3+\widetilde{\omega}^4}$ | $\dfrac{\widetilde{\omega}^3}{\widetilde{\omega}^1+\widetilde{\omega}^2+\widetilde{\omega}^3+\widetilde{\omega}^4}$ | $\dfrac{\widetilde{\omega}^4}{\widetilde{\omega}^1+\widetilde{\omega}^2+\widetilde{\omega}^3+\widetilde{\omega}^4}$ |
|---|---|---|---|---|
| 0 | (0.189, 0.624) | (0.164, 0.576) | (0.088, 0.400) | (0.076, 0.336) |
| 0.2 | (0.228, 0.569) | (0.194, 0.524) | (0.109, 0.358) | (0.090, 0.297) |
| 0.4 | (0.267, 0.514) | (0.224, 0.472) | (0.129, 0.317) | (0.103, 0.259) |
| 0.6 | (0.306, 0.459) | (0.255, 0.420) | (0.150, 0.275) | (0.117, 0.221) |
| 0.8 | (0.346, 0.404) | (0.285, 0.367) | (0.170, 0.233) | (0.131, 0.183) |
| 1 | (0.349, 0.349) | (0.315, 0.315) | (0.191, 0.191) | (0.145, 0.145) |

According to Equations (12) and (13), $\lambda$ cut sets of evaluation factors in Table 6 and fuzzy weight heavy $\lambda$ cut sets of evaluation factors in Table 7 are further calculated, and $\lambda$ cut sets of fuzzy scoring coefficients of each system of CNC punch loading and unloading truss robot are obtained. Then, the centroid value of the fuzzy scoring coefficient of the loading and unloading truss robot for CNC punch is obtained by defuzzifying the fuzzy number. Finally, the electrical control system is taken as the benchmark, and the scoring coefficient of the electrical control system is set as 1, and then the scoring coefficient of the mechanical system and the pneumatic system are calculated based on this, and the MTBF values of each system of the loading and unloading truss robot for CNC punch are finally obtained. The specific data are shown in Table 8, and the frequency diagram of the fault location is shown in Figure 5.

**Table 8.** The expected MTBF of each system of the loading and unloading truss robot for CNC punch.

| System | Score Centroid Value | Scoring Factor | MTBF Value (h) | Failure Rate (1/h) |
|---|---|---|---|---|
| Mechanical system | 1.840 | 0.867 | 2756.319 | $3.628 \times 10^{-4}$ |
| Pneumatic system | 1.797 | 0.847 | 2691.464 | $3.715 \times 10^{-4}$ |
| Electrical control system | 2.122 | 1.000 | 3179.145 | $3.146 \times 10^{-4}$ |

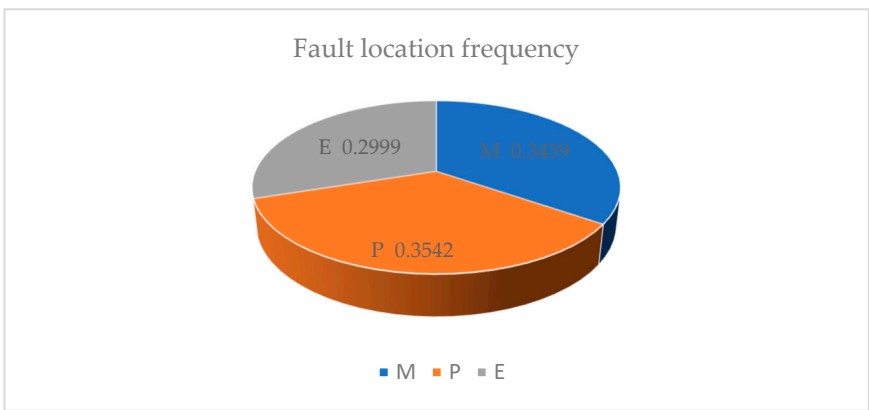

**Figure 5.** The frequency diagram of the fault location of the loading and unloading truss robot for CNC punch.

According to Table 8, the total failure rate of the loading and unloading truss robot of the CNC punch press is $1.049 \times 10^{-3}$ (1/h), that is, the total MTBF value is 953.289 h.

*3.3. Fault Location Analysis of FMEA*

Due to the influence of human factors on the reliability prediction method of the loading and unloading truss robot for CNC punch proposed in this paper, it is necessary to verify the effectiveness of the method and the accuracy of the results. Reliability data is the basis of FMEA. Only the real reliability data can make the reliability analysis results more accurate and ensure the practical significance of reliability work. The reliability data of the loading and unloading truss robot for CNC punch comes from the maintenance and after-sales department of a manufacturing enterprise of the loading and unloading truss robot for CNC punch. Through collecting and sorting out the fault maintenance reports of the loading and unloading truss robot for CNC punch for the whole year of 2022, the fault data are extracted for FMEA analysis.

Fault location analysis is to classify and count the fault location of the system or component of the product, find out the location with higher fault rate, and then find out the weak link of the product. Because the main research goal of this paper is to verify the reliability prediction results of the loading and unloading truss robot for CNC punch, only the fault location analysis is carried out.

Analyze the collected fault data according to the reliability model of the loading and unloading truss robot for CNC punch, and finally obtain 109 fault data of the loading and unloading truss robot for CNC punch. Then, failure location analysis is performed based on the above failure data. By analyzing the fault data of the loading and unloading truss robot of the CNC punch, the fault location frequency table and the fault location frequency diagram of the loading and unloading truss robot of the CNC punch are obtained, which are shown in Table 9 and Figure 6, respectively. Among them, the program error does not belong to the mechanical system, nor does it belong to the pneumatic system and the electrical control system, so it is listed separately, and the fault code is C [23].

**Table 9.** Frequency table of the fault location of the loading and unloading truss robot for CNC punch.

| Fault Location Code | Fault Location | Fault Record | Number of Failures | Total System Failures | Failure Frequency |
|---|---|---|---|---|---|
| M | Mechanical system | The frame of the raw material car is deformed | 1 | 34 | 0.3119 |
| | | X-axis chain is disconnected | 1 | | |
| | | U-axis drag chain collision parts | 1 | | |
| | | Clamp interferes with locating plate | 1 | | |
| | | Brush table slider damaged | 1 | | |
| | | X-axis motor is damaged | 1 | | |
| | | X-axis slider is damaged | 1 | | |
| | | Motor damage of raw material car | 1 | | |
| | | The brush machine motor is crushed by the panel | 1 | | |
| | | X-axis chain slot is damaged | 1 | | |
| | | Hydraulic elevator reducer oil leakage | 1 | | |
| | | The motor of the brush stand is burnt out | 1 | | |
| | | U-shaft unloading clamp reducer is damaged | 1 | | |
| | | U-shaft unloading clamp reducer does not turn | 1 | | |
| | | Z-axis reducer oil leakage | 1 | | |
| | | Z-axis is disconnected from the gear shaft connected with the reducer | 1 | | |
| | | Unloading reducer oil leakage | 1 | | |
| | | The wheel of the raw material car slipped | 1 | | |
| | | Fence deformation | 1 | | |
| | | U-shaft reducer oil leakage | 1 | | |
| | | The chain of X-axis is loose | 1 | | |
| | | U-axis chain is disconnected | 1 | | |
| | | X-axis chain groove deformation | 1 | | |
| | | Cylinder leakage | 1 | | |
| | | X-axis reducer oil leakage | 1 | | |
| | | The key between the U-axis motor and the reducer slips out | 1 | | |
| | | X-axis motor is burnt out | 1 | | |
| | | X-axis chain deformation | | | |
| | | The roller bearing of the raw material car is broken | 1 | | |
| | | Brush table roller slip | 1 | | |
| | | Raw material car motor abnormal sound | 1 | | |
| | | The U shaft interferes with the tooling cylinder during unloading | 1 | | |
| | | Brush table slip when full load | 1 | | |
| | | Motor coupling of raw material car falls off | 1 | | |
| P | Pneumatic system | Individual tooling sucker does not inhale | 1 | 38 | 0.3486 |
| | | The tooling cylinder rod is broken | 8 | | |
| | | Vacuum generator failure | 3 | | |
| | | Individual sucker cannot absorb the sheet material | 1 | | |
| | | The tooling cylinder broke the sucker | 1 | | |
| | | The reducing valve is damaged | 2 | | |
| | | Sucker failure | 2 | | |
| | | Tooling cylinder damage | 4 | | |
| | | Vacuum pump damage | 2 | | |
| | | Three-way pilot solenoid valve is damaged | 2 | | |
| | | The vacuum generator is damaged by oil | 1 | | |
| | | The vacuum generator is damaged by oil | 1 | | |
| | | The sucker has no suction | 1 | | |
| | | The tooling sucker is knocked off | 1 | | |
| | | Tooling cylinder positioning plate is not accurate | 1 | | |
| | | The gear cylinder does not move | 1 | | |
| | | The tooling sucker is damaged | 1 | | |
| | | Sucker with plate | 1 | | |
| | | Right-angle gas pipe joint leaks | 1 | | |
| | | Three-way gas pipe joint leakage damage | 1 | | |
| | | The tooling cylinder does not pop out sensitively | 1 | | |
| | | The tooling cylinder does not operate | 1 | | |

**Table 9.** *Cont.*

| Fault Location Code | Fault Location | Fault Record | Number of Failures | Total System Failures | Failure Frequency |
|---|---|---|---|---|---|
| E | Electrical control system | The curtain of light flickers frequently | 1 | 28 | 0.2569 |
| | | Proximity switch failure | 1 | | |
| | | Breaker damage | 1 | | |
| | | Displacement sensor failure | 1 | | |
| | | The emergency stop button is damaged | 1 | | |
| | | PLC controller damaged | 1 | | |
| | | Z-axis motor line is disconnected | 1 | | |
| | | Inverter damage | 3 | | |
| | | Hydraulic lifting platform towing chain line disconnected | 1 | | |
| | | Motor line of raw material car is disconnected | 1 | | |
| | | X-axis servo motor drive burned out | 1 | | |
| | | The indicator light of the electric cabinet is not on | 1 | | |
| | | The outer wiring of the light curtain is burnt by short circuit | 1 | | |
| | | The fan of the electric cabinet is damaged | 1 | | |
| | | Curtain failure | 1 | | |
| | | Inspection proximity switch is damaged | 1 | | |
| | | X-axis servo drive is damaged | 1 | | |
| | | The fan of the electric cabinet does not work | 1 | | |
| | | The loading drag chain is disconnected | 1 | | |
| | | Z-axis servo motor drive is damaged | 2 | | |
| | | Proximity switch failure | 1 | | |
| | | Full load proximity switch of hydraulic lifting platform is damaged | 1 | | |
| | | Access switch of U-axis retaking clamp is damaged | 1 | | |
| | | Unloading tow line is damaged | 1 | | |
| | | The light curtain line is broken by the groove cover | 1 | | |
| C | Program | The sucker failed to pick up the sheet | 1 | 9 | 0.0826 |
| | | Tooling cylinder interference plate | 1 | | |
| | | Sucker with plate | 1 | | |
| | | When loading, the sucker cannot absorb the sheet material alarm | 1 | | |
| | | Misalignment of loading material | 1 | | |
| | | U-shaft motor will alarm when unloading | 1 | | |
| | | Loading position is unstable | 1 | | |
| | | Loading rebound causes abnormal positioning | 1 | | |
| | | Z-axis motor often alarms | 1 | | |

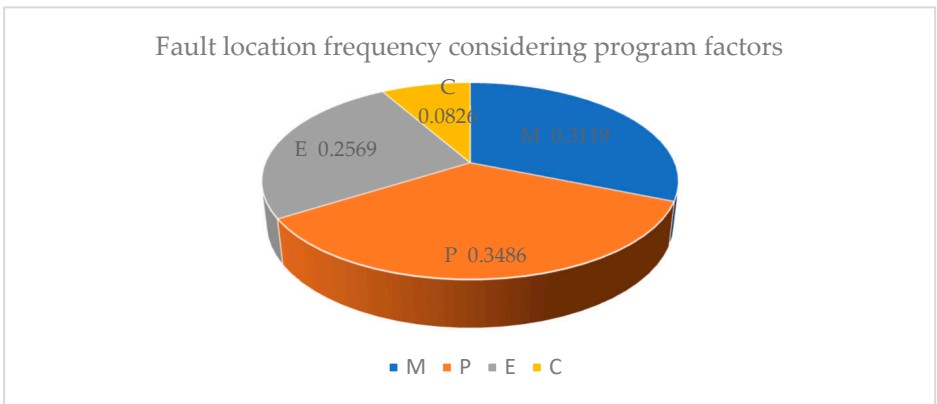

**Figure 6.** The frequency diagram of fault position of the loading and unloading truss robot for CNC punch considering program factors.

## 4. Discussion

The reliability prediction results show that the failure rate of the loading and unloading truss robot for CNC punch pneumatic system is $3.715 \times 10^{-4}$ (1/h), accounting for 35.42% of the whole system. The failure rate of mechanical system was $3.628 \times 10^{-4}$ (1/h), accounting for 34.59% of the whole system. The failure rate of electrical control system is $3.146 \times 10^{-4}$ (1/h), accounting for 29.99% of the whole system. Therefore, it is predicted that the pneumatic system is the weak link of the loading and unloading truss robot for CNC punch, which needs to be paid more attention.

After analyzing the fault data of the loading and unloading truss robot for CNC punch collected from the fault location, it can be seen from Table 9 and Figure 4 that the pneumatic system failure rate of the loading and unloading truss robot for CNC punch is the highest, accounting for 34.86% of the whole. Mechanical system failure rate followed, accounting for 31.19% of the total; electrical control system failure rate is low, accounting for 25.69% of the total; the failure rate of program problems is the lowest, accounting for 8.26% of the total. Therefore, it is concluded that the pneumatic system is the weak link of the loading and unloading truss robot for CNC punch.

Without considering the program, the failure rate percentage results of the pneumatic system, mechanical system, and electrical system of the loading and unloading truss robot for CNC punch are basically consistent with the reliability predicted results, so the verification method we adopted is effective, which verifies the effectiveness of the reliability predicted method and the accuracy of the predicted results.

## 5. Conclusions and Prospect

### 5.1. Conclusions

(1) Using reliability engineering and fuzzy theory, the reliability of the loading and unloading truss robot for CNC punch is studied, and a new reliability prediction method is proposed. Firstly, the failure rate of existing components is used, and the failure rate of an electrical control system is estimated to be $3.146 \times 10^{-4}$ (1/h) by the component counting method. Secondly, based on the predicted results of electrical control system, the failure rates of the mechanical system and pneumatic system are estimated to be $3.628 \times 10^{-4}$ (1/h) and $3.715 \times 10^{-4}$ (1/h), respectively, by using fuzzy theory. Finally, it is estimated that the failure rate of the loading and unloading truss robot for CNC punch is $1.0489 \times 10^{-3}$ (1/h), and the weak link is the pneumatic system. Therefore, the reliability prediction method proposed in this paper not only makes effective use of the existing reliability data of electrical control system, but also fully considers the actual characteristics of the lack of data of the mechanical system and pneumatic system. Combined with the project experience of relevant experts on the loading and unloading truss robot for CNC punch, the accuracy and operability of reliability prediction are improved;

(2) Due to the influence of human factors in the proposed reliability prediction method, the failure frequency, failure mode and main failure causes of each system of the loading and unloading truss robot for CNC punch are calculated through FMEA analysis on the collected fault record data of the loading and unloading truss robot for CNC punch. Among them, the pneumatic system is the most frequent failure system of the loading and unloading truss robot for CNC punch, accounting for 34.86%. Therefore, it is concluded that the pneumatic system is the weak link of the loading and unloading truss robot for CNC punch, which verifies the effectiveness of the reliability prediction method and the accuracy of the predicted results;

(3) The reliability prediction method proposed in this paper is consistent with the results of FMEA analysis based on actual maintenance fault record data, which indicates that FMEA analysis can be used to verify the reliability prediction results and provides a new research idea for the combination of reliability prediction and FMEA method.

### 5.2. Prospect

Although the reliability research of the loading and unloading truss robot for CNC punch has obtained good initial results, the reliability work of enterprises has just started, and the perfect reliability database has not been established. In the future, under the background of intelligent manufacturing, industrial network integration can be carried out on all equipment of the entire intelligent sheet metal automatic production line, real-time monitoring and comprehensive collection of all equipment fault information can be carried out, and a reliability database of the intelligent sheet metal automatic production line can be gradually established to achieve a wider range of automatic data flow in a wider

range of fields. The goal is to finally establish open, coordinated and shared reliability database management system. Then, the reliability research based on reliability database management system will get greater achievements.

**Author Contributions:** Conceptualization, K.Z.; writing, K.Z.; methodology, Z.J. (Zhixin Jia); writing—original draft preparation, R.B.; data curation, R.B.; supervision, Z.J. (Zhixin Jia); project administration, K.H.; funding acquisition, K.H.; investigation, Z.J. (Zhicheng Jia). All authors have read and agreed to the published version of the manuscript.

**Funding:** This research was funded by the Hainan Province Natural Science Foundation (Grant number 520MS061), National Key Research and Development Program of China (Grant Number: 2020YFB1709101).

**Institutional Review Board Statement:** Not applicable.

**Informed Consent Statement:** Not applicable.

**Data Availability Statement:** Not applicable.

**Acknowledgments:** This work was carried out with the support of the original breakdown maintenance record data provided by Yangzhou HENGA Automation Equipment Co. Ltd.

**Conflicts of Interest:** The authors declare no conflict of interest.

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
