# Peer review of "Reliability Prediction and FMEA of Loading and Unloading Truss Robot for CNC Punch"

_applsci, doi:10.3390/app13084951_

Round 1

Reviewer 1 Report

The FMEA method was used to analyze the fault position of the loading and unloading truss robot for CNC punch in the paper. The manuscript is more like a technical report than a scientific paper. There, as far as reviewers are concerned, is a lack of theoretical innovation. And moreover, the following points need to be clarified to improve the quality and the readability of the manuscript.  

(1) Is there any innovative work in the paper? What is the novelty of this paper with respect to the state of art? It is not easy to assess which is the main contribution of the work with respect to the present literature.

(2) The section “2. Materials and Methods” is the focus of the paper, but it lacks substantive content.

(3) What is the relationship between section 3.2 and 3.3, and please give the reason for the added “Program”?

(4) How to get the data in the tables, such as table 1, table 4, and so on?

(5) The abbreviation for failure mode and effect analysis FMEA should appear on line 65.

(6) The paper indicates that the pneumatic system is the weak link of the loading and unloading truss robot for CNC punch, but the conclusion doesn't accord with my cognition. The failure rate of the electrical control system, as far as I am concerned, should be the biggest. And moreover, this leads to a serious problem that only numerical simulation and results are presented in the manuscript, and the results seem interesting but some experiments should be used to verify the obtained results.

(7) The references of the manuscript, whose majority are dissertations, should be revised to emphasize the innovative work of the paper.

(8) At last, extensive editing of English language and style required.

Author Response

Thank you very much for your valuable advice, which is very valuable for me to revise this article and has played a great role in improving the technical level and readability of the article.

(1)The originality of this paper lies in the proposed reliability prediction method of the loading and unloading truss robot for CNC punch by using component counting method and fuzzy theory. The failure rate and MTBF value of electric control system are predicted by using component counting method. Secondly, based on the predicted results of the electrical control system, the expert scoring method based on the fuzzy theory is used to predict the reliability of the mechanical system and the pneumatic system, and the failure rate and MTBF values are obtained. Finally, the failure rate and MTBF value of the loading and unloading truss robot for CNC punch are predicted, and the weak link is analyzed. In order to verify the reliability of this method, this paper calculates the failure frequency, failure mode and main failure reasons of each system of the loading and unloading truss robot for CNC punch through FMEA analysis of the collected loading and unloading truss robot for CNC punch fault data, and obtains the weak link. In order to verify the validity of the reliability prediction method and the accuracy of the predicted results.

(2)This chapter for reliability prediction method and FMEA analysis method are explained, and according to your suggestions, added 2.3 chapter, to explain the reliability prediction method and FMEA analysis of the combination of the necessity and operation method, and give the overall implementation of the flow chart.

(3)The content in 3.2 is to use the proposed new reliability prediction method to predict the failure rate and weak links of the loading and unloading truss robot for CNC punch. In order to verify the effectiveness of this method and the accuracy of the results, FMEA is used to analyze the fault data provided by the manufacturer. According to the fault data analysis, failure causes include mechanical, electrical, and pneumatic system errors, as well as procedural errors, which are listed separately. In order to make the context of the article clearer, the title of Section 3.3 is now changed to fault location analysis of FMEA .

(4)The failure rate of the components is provided by the supplier, which is multiplied by the number of components, which is the failure rate shown in the fifth column of Table 1. After summing up, the MTBF value of the electrical control system can be obtained through the formula MTBF=1/λ. For the data in Table 4, more detailed determination methods and pictures have now been given in the manuscript.

(5)Abbreviations have been added.

(6)In this paper, a new reliability prediction method is proposed, and then FMEA analysis of the actual collection of fault maintenance record data, these data sources are real and reliable, so the reliability prediction method proposed in this paper is supported by practical and experimental verification; Secondly, the components used by the loading and unloading truss robot for CNC punch studied in this paper are imported from abroad and their quality is good, so the final result is that the pneumatic system is the weak link of the loading and unloading truss robot for CNC punch.

(7)The innovation of this paper is to propose a reliability prediction method for the loading and unloading truss robot for CNC punch by using component counting method and fuzzy theory. Moreover, FMEA is used to analyze the actual fault maintenance record data to verify the effectiveness of the reliability prediction method and the accuracy of the results. In the process of predicting the failure rate of mechanical system and pneumatic system, It also uses the expert scoring method based on fuzzy theory, establishes the fuzzy language set of scoring factors, and then uses analytic hierarchy process to determine the weight of scoring experts, and then solves the model of system scoring coefficient based on fuzzy theory.

(8)Some wording in the article has been revised to meet your standards.

Reviewer 2 Report

Reviewer’s Comments on applsci- 2289954

Reliability prediction and FMEA of loading and unloading 2 truss robot for CNC punch

This paper presents an approach to obtain the MTBF and the failure rate for a truss robot for CNC punch. The authors developed the approach based on the expert scoring method and fuzzy theory. Furthermore, the FMEA was considered, and a reliability assessment is provided for the case study. All in all, I consider that this is an interesting manuscript. I have the following comments:

1.     In the aims of improving the presentation of the proposed method, the authors are encouraged to provide a flow chart that summarizes all of the proposed steps, from the reliability modeling and assessment to the development of the FMEA and the final estimation. In general, please extent the content of the materials and methods section.

2.     Please discuss how the failure rates from table 1 were obtained, the authors have omitted the details here.

3.     In line 210, the authors mention “According to the fuzzy reliability prediction model [22]”, but no details of this approach have been provided. The authors are encouraged to discuss the characteristics of this model and to discuss what characteristics of the case make this model appropriate.

4.     Please provide more details about how the expert weights in Table 4 were determined.

5.     The authors are encouraged to provide insights for future research in the conclusion section.

Author Response

(1)Thank you very much for your valuable advice, which is very valuable for me to revise this paper and has played a great role in improving the technical level and readability of this paper. The overall method flow chart of this paper has been added in Chapter 2.3 of the manuscript, and the content of materials and methods has also been expanded.

(2)The failure rate of the components is provided by the supplier, which is multiplied by the number of components, which is the failure rate shown in the fifth column of Table 1. After summing up, the MTBF value of the electrical control system can be obtained through the formula MTBF=1/λ.
(3)Because the mechanical system and pneumatic system of the loading and unloading truss robot for CNC punch include a large number of parts and components, the failure mechanism of the parts and components of the system is different. On the one hand, the reliability prediction work of the loading and unloading truss robot for CNC punch has just been preliminarily carried out. There is no reliability prediction data for the mechanical system and pneumatic system that can be directly used. Moreover, there are a large number of homemade parts in the mechanical system, and the standardization degree is low, so the failure rate of homemade parts cannot be directly checked through the reliability prediction manual. On the other hand, the occurrence of failures of many components of pneumatic system is highly related to pipeline connections and auxiliary components, so if the reliability level is predicted simply by the method of counting components, there will often be a large error. On the other hand, the failure rate of mechanical and pneumatic systems is also closely related to a series of uncertain factors such as manufacturing, assembly, use and maintenance. Considering the above reasons, this paper adopts the expert scoring method based on fuzzy theory to predict the reliability of mechanical system and pneumatic system of the loading and unloading truss robot for CNC punch. The specific operation method of this model is to first establish the fuzzy language set of scoring factors, then determine the weight of scoring experts, and then solve the model based on the fuzzy theory of the system scoring coefficient, and detailed steps are described in the manuscript.

(4)More detailed methods of determination and pictures have now been given in the manuscript.

(5)In the newly revised manuscript, views on future research are added in the conclusions and prospects of section 5.

Round 2

Reviewer 1 Report

The revised manuscript has been a great improvement. The reply provided by the authors, however, is difficult to trace back. Meanwhile, the figure 1 in “applsci-2289954-coverletter” is confusing, and this shows that the author is not serious enough. Moreover, the author did not provide evidence of the authenticity of the data shown in table1, table 4, and so on. In this case, the results and conclusions provided by the authors should be verified by experiments.

Author Response

    Thank you very much for your advice. After careful consideration, I found the flow chart was not clear enough, so I revised it again.
    The data collection method in Table 1 is as follows: Calculate the overall failure rate by collecting the name, model, quantity, and failure rate of each component. The failure rate of electronic components is estimated based on GB/T 37963-2019 "Electronic Equipment Reliability Prediction Model and Data Manual" and GJB 299-87 "Electronic Equipment Reliability Prediction Manual" and the failure rate of components provided by the supplier.
In Table 4, due to the subjective factors of each expert will be mixed with the independent scoring method of experts, in order to eliminate the human influence, it is necessary to adopt the analytic hierarchy process, so as to obtain the expert weight, and then calculate the evaluation value of each expert based on the fuzzy theory according to the weight of each expert.
    For you think the need to have experimental evidence, the FMEA analysis method used in this article is through the actual collection of fault maintenance record data analysis to draw the conclusion of the weak link, the FMEA analysis method has been equivalent to the experimental verification.

Round 3

Reviewer 1 Report

The comments have been dealt with in the revised manuscript. However, the reviewer disagreed with the author's point of view which the FMEA analysis method is equivalent to the experimental verification, because the authors did not give reasons for the authenticity of the data in the tables in the manuscript. And therefore, the results and conclusions provided by the authors, as far as I'm concerned, must be verified by experiments in this case.